# Immunological Misfiring and Sex Differences/Similarities in Early COVID-19 Studies: Missed Opportunities of Making a Real IMPACT

**DOI:** 10.3390/cells12222591

**Published:** 2023-11-08

**Authors:** Aditi Bhargava, Johannes D. Knapp

**Affiliations:** 1Center for Reproductive Sciences and Department of ObGyn, University of California San Francisco, San Francisco, CA 94143, USA; 2Aseesa Inc., Hillsborough, CA 94010, USA; jk@aseesa.com

**Keywords:** CD4Temra, GzB^+^CD8, hydroxychloroquine, ICU, IL-1β, IL-18, IFN, obesity, pDCs, Remdesivir, sex differences, T cells, Tocilizumab

## Abstract

COVID-19-associated intensive care unit (ICU) admissions were recognized as critical health issues that contributed to morbidity and mortality in SARS-CoV-2-infected patients. Severe symptoms in COVID-19 patients are often accompanied by cytokine release syndrome. Here, we analyzed publicly available data from the Yale IMPACT cohort to address immunological misfiring and sex differences in early COVID-19 patients. In 2020, SARS-CoV-2 was considered far more pathogenic and lethal than other circulating respiratory viruses, and the inclusion of SARS-CoV-2 negative patients in IMPACT cohorts confounds many findings. We ascertained the impact of several important biological variables such as days from symptom onset (DFSO); pre-existing risk factors, including obesity; and early COVID-19 treatments on significantly changed immunological measures in ICU-admitted COVID-19 patients that survived versus those that did not. Deceased patients had 19 unique measures that were not shared with ICU patients including increased granzyme-B-producing GzB^+^CD8^+^ T cells and interferon-γ. Male COVID-19 patients in ICU experienced many more changes in immunological and clinical measures than female ICU patients (25% vs. ~16%, respectively). A total of 13/124 measures including CCL5, CCL17, IL-18, IFNα2, Fractalkine, classical monocytes, T cells, and CD4Temra exhibited significant sex differences in female vs. male COVID-19 patients. A total of nine measures including IL-21, CCL5, and CD4Temra differed significantly between female and male healthy controls. Immunosuppressed patients experienced the most decreases in CD4Temra and CD8Tem cell numbers. None of the early COVID-19 treatments were effective in reducing levels of IL-6, a major component of the cytokine storm. Obesity (BMI >30) was the most impactful risk factor for COVID-19-related deaths and worst clinical outcomes. Our analysis highlights the contribution of biological sex, risk factors, and early treatments with respect to COVID-19-related ICU admission and progression to morbidity and mortality.

## 1. Introduction

SARS-CoV-2, the virus that causes COVID-19, can cause pneumonia, acute respiratory distress syndrome (ARDS), and death [1,2]. SARS-CoV-2 infection causes a mild-to-moderate illness in the majority of infected individuals despite direct exposure [3]; in a subset of individuals, these unremarkable symptoms can suddenly develop into severe disease, requiring hospitalization, oxygen support, and/or admission to an intensive care unit (ICU) [1,2]. Because SARS-CoV-2 has an unusually long incubation period, ranging from 2 to 14 d, prolonged presence of the virus in the respiratory tract up to a month after initial infection [3,4] may explain this sudden turn of events. Development of cytokine storm in a subset of patients with severe COVID-19 illness along with impaired gas-exchange function is thought to result in ARDS, multi-organ failure, and death [5,6].

COVID-19 symptoms overlap with other well-characterized viral infections, including the flu; thus, it is important that data reported for COVID-19 patients does not include patients that are in fact SARS-CoV-2 (COVID-19)-negative. Several risk factors have been identified that determine the progression of mild COVID-19 to a severe and critical stage. These risk factors include, but are not limited to, old age; male sex; pre-existing comorbidities such as chronic lung, heart, liver, and kidney diseases; hypertension, diabetes, obesity, tumors, immunodeficiencies, and pregnancy [7]. COVID-19-related complications include acute kidney injury, coagulation disorders, and thromboembolism. COVID-19 disease progression is monitored using laboratory parameters such as lactate dehydrogenase, procalcitonin, C-reactive protein, and proinflammatory cytokines such as interleukin (IL)-6, IL-1β, Krebs von den Lungen-6 (KL-6), and ferritin [7,8]. Predictive and machine learning models such as the artificial-intelligence-based chi-square automatic interaction detection (CHAID) method and LogNNet neural networks have been used to predict biomarkers and decision trees for COVID-19 patients [9,10]. A high degree of accuracy was achieved in predicting diagnosis of COVID-19 disease using these models [9,10]. However, for these models to be accurate and reproducible, it is imperative that multiple datasets are used for training and that the datasets are reliable and reproducible.

Interferon (IFN)-mediated antiviral responses precede pro-inflammatory ones, optimizing host protection and minimizing collateral damage [11,12]. Deviations from these balanced responses can be detrimental. After SARS-CoV-2 infection, studies have shown that IFN-λ and type I IFN production are delayed, dampened, and induced in severely ill patients. Thus, early, mid, and late immune responses from days after symptom onset (DFSO) are critical for understanding the immunopathology of COVID-19. IL-18 and IL-1β are secreted in response to SARS-CoV-2-activated inflammasome and, left unchecked, can lead to lytic cell death or pyroptosis [13]. IL-18 is identified as one of 19 cytokines to stratify hospitalization and mortality in COVID-19 patients [14]. IL-18 levels are influenced by sex hormones and age and increased in metabolic diseases, including diabetes and obesity [15].

Data from COVID-19 patients when several early treatments were being tested are highly valuable. The Yale IMPACT cohort is one such study that collected clinical data and biological samples from COVID-19 patients that were hospitalized in early 2020 [16,17]. The patient demographics and anthropomorphic features of the IMPACT cohort have been described in the original articles [16,17]. The IMPACT cohort data reported in two *Nature* studies [16,17] are confounded by the inclusion of several SARS-CoV-2-negative patients. In addition, there were many important biological variables in the IMPACT cohort that were not considered in the data presented in the original articles [16,17]. The patients included in the Yale IMPACT cohort had considerable variability in days from symptoms onset (DFSO); the range was days 1 to day 47. The immune profile in the early phase of infection is expected to differ vastly from mid-to-late phases of infection, which likely coincides with recovery or is the tipping point for progression to severe illness; thus, it is important to delineate changes in immune profile with respect to DFSO. The disease severity score (clinical score) in this cohort ranged from 1–5. Some patients were in ICU when first enrolled whereas others were not. The patients fell into at least five different categories of risk factors for COVID-19, whereas many patients ill with COVID-19 did not have these risk factors. Thus, non-ICU COVID-19 patients who also had many of the risk factors as those in ICU would be a pertinent comparison group in addition to healthy healthcare workers (HCW).

The IMPACT cohort patients were given at least four different treatments for COVID-19 with one hundred thirty-two and one hundred sixty-one of one hundred seventy-nine patient datapoints having received Tocilizumab (Toci) and hydroxychloroquine (HCQ), respectively, regardless of disease severity or underlying risk factors; yet, the impact of these treatments on immune measures remains to be evaluated. Corticosteroids and Toci are approved treatment for cytokine-release syndrome (cytokine storm), but the study failed to evaluate whether any of these treatments were effective in reducing levels of target proinflammatory cytokines.

Sex differences in several immune measures such as IL-18, IL-8, and T cell numbers in the IMPACT cohort are reported [17]. In this study, we tested two hypotheses: (1) Obese patients (BMI > 30) experienced more dramatic changes in immunological measures than those with other risks; (2) Few sex differences would exist in immune measures at baseline in healthy controls and novel sex differences, albeit not many, would appear after disease onset. We also aimed to determine the distinct immune signature of deceased patients not shared with ICU and non-ICU COVID-19 patients that might put them at greater risk for mortality and severe adverse outcomes. The impact of key variables such as DFSO, risk factors including obesity (BMI ≥ 30), treatments received, treatment counts, clinical score, biological sex, and ICU status on the significantly changed immunological signature of deceased, non-ICU, and ICU female and male patients was evaluated.

## 2. Materials and Methods

### Statistical Methods and Data Analysis

IMPACT Yale cohort data (Table 41586_2020_2700_MOESM1_ESM) was used for analysis in this report [16,17]. We included samples with confirmed viral load (either nasopharyngeal (Np) or saliva) and excluded samples with unavailable or zero measurements for both Np and saliva load. We first ascertained immune measures unique to deceased patients versus HCW. Next, we ascertained changed immunological measures in female and male SARS-CoV-2^+^ ICU patients using non-ICU SARS-CoV-2^+^ female and male patients, respectively, as comparison groups. Next, we determined how changed immunological measures in deceased and ICU patients were affected by biological variables present in the cohort such as DFSO, risk factors, early treatments, and clinical scores. Here, we show that careful analyses using different groups is essential for understanding complex datasets that contain biological variables that cannot always be defined or accounted for. Obesity is a significant risk factor for COVID-19 and adjusting for BMI can be misleading. Finally, sex-aggregated analysis can be misleading, especially for measures that are significantly changed in opposite directions between males and females. We used Aseesa Stars (www.aseesa.com; accessed on 20 July 2023) for all analysis performed as described below and for the generation of all charts and graphs.

Box–Cox transformations were applied to every test/control group pair with at least 5 samples present in both groups in order to stabilize variance and improve normality. Transformed values were only used for significance tests and not shown in the charts. Optimal exponents (λ) were estimated as described in [18], using the normal identity line in place of normality tests. Exponents between −5 and 5 were considered and tested at increments of 0.00015625 within the interval (−0.015, 0.015) or at increments of 0.005 otherwise. For every tested exponent, a standardized normal quantile–quantile (QQ) plot was constructed for both test and control, and the average distance from the normal identity line was calculated. The exponent with the greatest average distance across test and control was used. λ was generally stable across test groups, even with unequal sample sizes.

Correlation Scatter Plots: Samples with a zero value for the query symbol (biological or clinical measure), or with an empty value for the second symbol were excluded from correlations. Linear regression curves, or an n^th^-degree polynomial regression curve if its goodness-of-fit is either 50% greater than or if it explains at least half of the variance not explained by the (n−1)^th^-degree polynomial, were fitted to the data. R^2^, r, and *p* denote the goodness-of-fit, Pearson’s/Spearman’s correlation coefficient, and significance of the correlation, respectively. Measures were treated as ordinal if all values were integers and if the number of unique values was less than sqrt(number of available values), namely risk factor count, COVID_RISK_(1–5), Ethnicity, Days from Remdesivir Start/End, (Active) treatment count, ICU, clinical score, Latest Outcome, and coagulopathy. Spearman’s and Pearson’s correlation coefficients were calculated for ordinal measures and for the remaining numeric measures, respectively.

Bar Charts and Heat Maps: Bar charts show log_2_ fold change versus control, with error bars denoting the standard deviation and the filled fraction of a bar denoting the percentage of samples with non-zero values. ***, **, and * denote *p* < 0.001, 0.01, and 0.05 versus control, respectively, whereas ^†††^, ^††^, and ^†^ denote *p* versus the preceding test group (one bar above), using Welch’s *t*-test. Degrees of freedom were calculated using the Welch–Satterthwaite equation, and exact *p*-values were calculated using the cumulative t-distribution functions as *p* = 2 * MIN(P(x), Q(x)). The comparison mode Value to Average was used for all relative (change versus) charts, such that the change $g=\bar{G}$ was calculated by averaging the set of all individual changes $G=\{x\in T:\log_{2}\left(x/\mu_{C}\right)\}$ versus the outlier-adjusted control group average $\mu_{C}$, for values x included in the test group T, with the standard deviation given by $\sigma \left(G\right)$. The control average was adjusted for outliers by calculating and applying the average individual change in each control sample versus the unadjusted control average. Absolute bar charts show the average and standard deviation for the control group and for all test groups. Labels in parentheses next to symbols show the average in the first control group, whereas labels in bars show the average in the respective test group. Labels next to test group names show the number of samples with non-zero values in the respective group. Additionally, up to five significantly correlated measures are shown in the chart legend, sorted by ascending *p*-value. Only samples included in the bar chart’s control and test groups with a non-zero value for the query symbol and a non-empty value for the second symbol/metadata characteristic are included in correlations. Heat maps provide a bird’s eye view of the changes allowing for simultaneous visualization of multiple measures rather than individual bar charts. The heat maps show log_2_-fold changes in all immunological parameters that were significant in ICU versus non-ICU female and male patients. The same measures were then analyzed for variables such as DFSO, clinical scores, etc. Values shown in heat maps are calculated in the same way as those in bar charts. Outliers in heat maps, defined as less than Q1 − 1.5(IQR) or greater than Q3 + 1.5(IQR), are highlighted in yellow. Interquartile ranges (IQRs) were calculated by interpolating between data points to determine empirical quantiles. Percentages next to symbols represent the percentage of samples in which the symbol was detected across all test groups. Numbers above test group labels show the number of samples that were included (n); if not all symbols were present in the same number of samples a range is shown.

Volcano Plots: Values shown in volcano plots are calculated in the same way as those in bar charts. The color intensity of points represents the size of the change, with increased symbols drawn in red and decreased symbols in blue. Green lines denote (from top to bottom) *p* < 0.001, 0.01, and 0.05 per the Welch’s *t*-test.

Principal Component Analyses (PCA): PCA was used for reducing the dimensionality of the dataset, thereby increasing interpretability and simultaneously minimizing information loss. PCA creates new uncorrelated variables that successively maximize variance by solving an eigenvalue/eigenvector problem; the new variables are defined using the dataset at hand, not a priori, hence making PCA an adaptive data analysis technique [19]. For Symbol PCAs, only symbols with *p* < 0.05 were included, and samples with zero values were excluded. Covariance matrixes were created by standardizing all values for each symbol using $z=\frac{v-\mu }{\sigma }$ for a sample value v, group average μ, and standard deviation σ, and calculating the covariance between two symbols. Eigenvalues and eigenvectors were calculated using the gsl_eigen_symmv function from the GNU Scientific Library [20]. The total dataset variance was calculated by summing the absolute eigenvalue for each symbol; component contributions were calculated by dividing each eigenvalue by the total dataset variance, and the correlation of individual symbols with a component was given by its eigenvector. Donut charts show the primary components necessary to explain at least 90% of the dataset’s variance, and the symbols correlated most with the first four primary components. PCA biplots show the correlation of each included symbol with Component 1 (*x*-axis) and Component 2 (*y*-axis) as given by the components’ eigenvectors. The color of points in biplots for Symbol PCAs denotes log_2_ fold change versus the control, calculated in the same way as in bar charts.

Venn Diagrams: Venn diagrams contrast two test groups, showing the number of exclusive significantly changed symbols in each group (*p* < 0.05 per the Welch’s *t*-test; left/right), and the number of shared significantly changed symbols in the same direction (middle) and in opposite directions (top). Additionally, the most changed exclusive symbols are shown to the left/right of each test group; shared significantly increased and decreased symbols are shown at the bottom left and bottom right, respectively, symbols that are significantly increased in the first and significantly decreased in the second group shown at the top left, and symbols that are significantly decreased in the first group and significantly increased in the second group are shown at the top right. The symbols shown were sorted by average rank across the sorting methods *p*-value, relative change, and absolute change.

## 3. Results

### 3.1. SARS-CoV-2 Viral Load and Its Correlation with Immunological Measures

SARS-CoV-2 viral load in saliva or nasopharyngeal swabs as detected via RT-PCR were used to diagnose confirmed COVID-19 cases at the time of hospital admissions and screening, irrespective of symptoms [16,17]. However, the screen data are not provided or available. In our analysis of the IMPACT cohort, we only included samples with at least one non-zero measurement of SARS-CoV-2 saliva or NP load. We found that viral load correlated with distinct cytokines/chemokines including IFN-γ, TNF-α, and CCL8 (Appendix A). The top four correlations for each viral load in saliva and nasopharyngeal swabs that were found are shown in all SARS-CoV-2^+^ patients and HCW. Saliva load correlated with CCL8 (r = 0.548, *p* = 10^−^^3.7^). CCL1 was highly correlated with several chemokines/cytokines including CCL21, CCL8, IL-10, and IL-6 (Appendix A). Nasopharyngeal (NP) viral load correlated negatively with AntiS1IgG (r = −0.418, *p* =10^−^^3.3^).

#### 3.1.1. Obesity Is a Risk Factor for COVID-19 Severity

In the IMPACT cohort, a subset of patients fell into one or more of five different categories of risk factors for COVID-19, namely cancer treatment during the past year, chronic heart disease, hypertension, chronic lung diseases, and immunosuppression [16]. Extreme BMIs (≥35) are correlated with an increased relative risk of mortality [17], yet the BMI was not considered as a risk factor; instead, the authors adjusted the data for BMI and age. Both female and male patients were considerably older than the HCW and the BMI was significantly higher in female COVID-19 patients than other groups (Figure 1a). Clinical scores are often predictors of health outcomes and used as a surrogate for disease severity. In our reanalysis, we found that deceased patients and those with a clinical score of 5 had the highest average BMI of 37 and 37.8, respectively, whereas HCW had an average BMI of 26.8 (Figure 1b). Of the ~140 reported biological and clinical measures in the IMPACT dataset, the BMI correlated most significantly with AntiS1-IgG levels and negatively with dendritic cells (DCs); the top five most significant correlations are shown (Figure 1b) and a full list of all correlations for all biological and clinical measures can be found in Appendix A.

In our analysis, we considered obesity (BMI ≥ 30) as an additional risk factor as we found that obese patients had the worst clinical score, followed by patients with chronic lung disease (Figure 1c). Immunosuppressed patients did not fare any worse than patients with other risks such as chronic heart disease, prior cancer treatment, or hypertension. Deceased patients and those in the ICU had the highest clinical scores (>4.0), whereas non-ICU SARS-CoV-2^+^ patients had an average clinical score of 1.75 (Figure 1c). Patients who were >28 over days from symptoms onset had the worst clinical score and those on corticosteroid treatment also had the worst score (Figure 1d), suggesting that none of the early treatments were effective. Regardless of risk, treatment, or DFSO, the BMI and clinical score correlated with cytokines/chemokines IL-6 and CCL1 and with treatment counts. These findings suggest that obesity/BMI should be classified as a major risk factor for COVID-19 health outcomes and should not be adjusted for.

#### 3.1.2. Immunological Profile of COVID-19 Patients in ICU and Those with Coagulopathy

It is expected that individuals infected with SARS-CoV-2 in ICU and those in critical conditions will have a hyper activated immune system. Not surprisingly, the original reports found many changes in immune cell numbers, T cell subsets, and cytokine/chemokine levels in COVID-19 patients compared with healthy HCW controls. Principal component analysis (PCA) scatter plots and donut charts for symbols revealed the contribution of the various components necessary to explain >90% of the dataset’s variance in non-ICU, ICU, deceased, and CAC patients versus HCW (Appendix A). Non-ICU patients had the most variability in the dataset (Appendix A). In patients with coagulopathy, due to low sample size only two components were sufficient to explain nearly 100% of the variability (Appendix A). T cells, CXCL10, IL-10, and IL-6 were amongst the most changed measures in ICU, deceased, and CAC patients when compared with HCW (Appendix A). Follicular CD8 T cells, CD8Tcm, pDCs, T cells, IL-6, CCL1, and IFN-γ were amongst the most changed measures when compared with non-ICU patients (Appendix A). Of note, all CAC and deceased patients were also in the ICU, but when segregated CAC and deceased/ICU patients shared only three measures; comparatively, ICU and deceased patients shared ten measures as compared with non-ICU patients (Appendix A). There were several shared measures that changed in opposite directions in subpopulations of ICU patients. For example, IL-6, IFN-γ, and CCL15 levels increased in the deceased but decreased in CAC patients, whereas IL-2+CD4 numbers increased in CAC but decreased in deceased patients. Similarly, GzB^+^CD8 and CD8 T cell numbers increased significantly in deceased but not ICU patients (Appendix A).

In sex-aggregated analysis, non-ICU and ICU patients had ~53% (67/127) and ~42% (54/127) significantly changed measures compared with HCW (Figure 2a–c). Of those changed measures, forty-nine were shared (either ↑ or ↓), eighteen were specific to non-ICU, and five to ICU patients as compared to HCW (Figure 2a). Non-ICU and deceased patients also shared 50 measures as all deceased patients were also in the ICU (Figure 2b). Deceased and ICU patients shared 44 measures, whereas 21 were significantly changed in deceased and 10 in ICU patients, respectively (Figure 2c). The top increased measures relative to HCW included T cells, plasmacytoid dendritic cells (pDCs), DC1, total monocytes (Mono), IL-8, IL-10, CXCL10, and IL-6 in deceased, non-ICU, and ICU patients as seen in the volcano plots (Figure 2d) and Appendix A. A total of 15 measures were significantly different in ICU versus non-ICU patients. TNF-β and CCL7 were also highly increased in deceased ICU patients, whereas NKT cell and CD8 T cell numbers decreased in ICU patients; but, in deceased ICU patients CD8 T cell numbers tended to increase, although they did not attain statistical significance (Figure 2e). In sex-segregated analysis, CD8 T cells were no longer significant (Figure 2e and Appendix A). Thus, in measures changed in opposite directions, the differences are lost when performing sex-aggregated analysis. In immunosuppressed patients, levels of TNF-β, FLT3L, and CCL7 tended to increase (Appendix A). FLT3L levels were increased >28 days (Appendix A). TNF-β and CCL7 were also most increased with a clinical score of 5 and in patients that received all four treatments or any given individual treatment (Appendix A).

#### 3.1.3. IL-2, GzB^+^CD8, and Naïve CD8 T Cells Are the Most Changed Measures in Deceased COVID-19 Patients

Deceased COVID-19 patients had 22 and 28 immune measures that were not shared with ICU and non-ICU patients, respectively (Figure 3a and Figure 4a). IL-2, GzB^+^CD8 effector cells, IL-1α, and CD8Temra were the most increased relative to HCW. The top decreased measures included naïve CD8 T cells and IL-2+CD4 cells in deceased patients versus HCW (Figure 3a). In female COVID-19 patients, levels of IFN-α2, CCL17, Fractalkine, CD4Temra, and naïve CD8 T cells were higher than in male COVID-19 patients (Figure 3b). CD4Temra cell numbers were higher in female HCW versus male HCW, whereas IL-1β levels tended to be lower in female HCW versus male HCW (Figure 3b). Most measures were also changed with a clinical score of 5 and >15 days from symptoms onset, consistent with ICU patients having >4 clinical scores. Immunosuppressed patients experienced the greatest decrease in CD4Temra and patients with prior cancer treatment experienced the greatest decrease in naïve CD8 T cell numbers compared with other risk factors (Figure 3c). Remdesivir was slightly better than other treatments in bringing down levels of several measures nearer to those of HCW (Figure 3c). GzB^+^CD8 cell numbers were most increased in deceased patients and inversely correlated robustly with CD4 T cell numbers in deceased (r = −0.79, *p* = 10^−^^3.6^) and non-ICU patients (r = −0.658, *p* = 10^−^^7.3^) but not in ICU patients (r = −0.101, *p* = 0.744). A positive correlation was also seen between GzB^+^CD8 and CD8 T cell numbers in deceased and non-ICU patients and with CXCL13 in only deceased patients (Figure 3d). GzB^+^CD8 and CD8Temra cell numbers were also correlated in all sub categories of ICU patients; CD4 and CD8 T cells exhibited a strong negative correlation, especially in non-ICU (r = −0.914, *p* = 10^−^^22.4^) and deceased (r = −0.918, *p* = 10^−^^6.3^) patients (Figure 3d).

#### 3.1.4. Vascular Endothelial Growth Factor-A (VEGF-A), IL-15, Eotaxin3, and CXCR3+CD8 Are the Most Changed Measures in Non-ICU COVID-19 Patients

Non-ICU COVID-19 patients experienced significant changes in 28 measures that were not significantly changed in deceased COVID-19 patients (Figure 3 and Figure 4a). VEGF-A, a secreted growth factor known to induce endothelial cell proliferation, promote cell migration, angiogenesis, induce permeabilization of blood vessels, and inhibit apoptosis, was highly increased in non-ICU and ICU COVID-19 patients compared with HCW. Eotaxins are chemotactic for eosinophils and induce their degranulation. Eotaxin-1 is thought to play a role in COVID-19-related neurologic syndrome [21]. In aggregated analysis, Eotaxin-3 and Eotaxin-1 levels decreased in COVID-19 patients; however, in sex-segregated analysis, Eotaxin-3, CCL5, IL-21, and IL6^+^CD4^+^ T cell levels were lower in female versus male HCW and this sex difference was maintained in female versus male COVID-19 patients (Figure 4b). Sex differences in the number of classical monocytes (cMono), epidermal growth factor (EGF), and Eotaxin-1 became apparent in female versus male COVID-19 patients (Figure 4b). A number of non-ICU-significant measures were also significantly increased in patients with lower clinical scores of 1 and 3 and in obese patients with a BMI > 30. VEGF-A levels were increased in all treatment groups with Remdesivir treatment increasing levels more than others (Figure 4c). VEGF-A levels were also highest in patients with DFSO >28 days and in patients that were immunosuppressed or had received cancer treatment (Figure 4c).

#### 3.1.5. Sex Differences and Similarities in Immunological Profile of COVID-19 Patients

To better understand what immunological measures contributed to observed differences in COVID-19 outcomes between female and male patients, we next focused our analysis on non-ICU and ICU patients alone (irrespective of outcome) as they were well-matched for clinical score (Figure 5a), age, treatment, and risk factor counts (Figure 1a). PCA donut charts show several components that explain >90% variance in female and male patients, respectively, with component 1 explaining 29% variability in the dataset in the females and 40% variability in the male patients (Figure 5b). PCA scatter plots depicts measures that contributed to the first two components (Figure 5b,c). Female ICU patients only had 20 significantly changed measures, whereas male ICU patients had 32 changed measures compared with non-ICU female and male patients, respectively (Figure 6a). A total of seven measures were significantly changed in the same direction; IL-6, IL-10, CCL1, and ncMono were increased and pDCs, T cells, and CD8Tcm were significantly decreased in ICU male and female patients compared with non-ICU male and female patients, respectively (Figure 6b). Of the shared measures in both sexes, IL-6 and IL-10 increased the most as the disease progressed (DFSO) or worsened (clinical score) and in patients with BMI > 30 and those with prior cancer treatment (Figure 6b,c). None of the treatments were effective in reducing levels of IL-6 or IL-10 and T cell numbers were higher in female versus male patients, but cell numbers did not differ in female and male HCW (Figure 6c).

#### 3.1.6. CXCL10 and PD1^+^TIM3^+^CD4^+^ T Cells Were the Most Increased and cDC2s and NKT Cells Were the Most Decreased Immunological Measures in Female ICU versus Non-ICU Patients

Of the significantly changed measures specific to female ICU patients versus non-ICU female patients, five measures were increased and seven were decreased (Figure 6b). TIM3 is considered to be an exhaustion marker for CD4^+^ T cells [22] and PD-1 is expressed on activated CD4+ and CD8+ T cells and on several other subpopulations of immune cells including B cells, NKT cells, and monocytes and provides an inhibitory signal to dampen T cell receptor signaling [23]. DC2s are ontogenically and functionally heterogeneous, with the presentation of exogenous antigens to CD4^+^ T cells for the initiation of T helper cell differentiation being ascribed as their main function [24]. Elevated CXCL10 levels are associated with severe COVID-19 and worse progression. CXCL10 is considered a predictive biomarker of patient outcome in COVID-19, and increased levels are associated with ARDS and neurological complications of COVID-19. PD1^+^TIM3^+^CD4^+^ T cells and CXCL10 were significantly increased in female ICU but not male ICU patients compared with non-ICU COVID-19 patients. In this analysis, CXCL10 levels and PD1^+^TIM3^+^CD4^+^ T cell numbers further increased as the disease score worsened and up to 7–10 days after symptoms onset. Comorbidities/risks such as chronic heart disease and hypertension also increased CXCL10 levels and none of the treatments were effective in reducing its levels to baseline (Figure 6c). DC2 cell numbers decreased further as the disease score worsened and up to 11–14 days after symptoms onset. Comorbidities/risks such as a BMI > 30, chronic heart disease, and hypertension also further decreased DC2 cell numbers and none of the treatments were effective in bringing the numbers back up to baseline (Figure 6c). CD4Temra and T cell numbers were significantly higher, whereas IL-18 levels were significantly lower in female versus male COVID-19 patients (Figure 6c and Figure 7b).

#### 3.1.7. IL-12p70 and IL-2 Were the Most Increased and CD8Tem and T cells Numbers Were the Most Decreased Immunological Measures in Male ICU Versus Non-ICU Patients

Of the significantly changed measures specific to male ICU patients versus non-ICU male patients, twenty=two measures were increased and three were decreased (Figure 6b and Figure 7a). IL-12 and IL-2 levels are increased in COVID-19 patients but their association with disease severity is controversial [25,26]. In this analysis, IL-12p70 and IL-2 levels were highest in male ICU patients and decreased in patients with a clinical score of 1 versus HCW (Figure 7a,b). T cell numbers decreased further as the disease score worsened and were the lowest in patients at 22–28 days after symptoms onset. All comorbidities/risks further decreased T cell numbers with prior cancer treatment having the biggest impact on T cell numbers. None of the treatments were effective in bringing the numbers back up to baseline (Figure 7b).

#### 3.1.8. Correlation of Key Cytokines and Immune Cell Types with Other Clinical and Immunological Measures

Cytokines such as IFN-γ and IL-6 are induced early on after viral infection and are key components of the “cytokine storm”. Plasmacytoid dendritic cells and monocytes are also key for fighting off infections as they produce several cytokines including IFNs. IFN-γ, IL-6, PD1^+^TIM3^+^CD4^+^ T cells, and ncMono were all increased in ICU female and male patients compared with HCW and non-ICU patients, whereas pDC numbers were highly reduced in ICU female and male patients (Figure 8a). For each cytokine/cell type, the five most correlated measures (sorted by *p*) are shown with their respective *p*-value and Pearson’s r-values in the bar charts. Correlations with clinical measures such as ICU, clinical scores, treatment counts, and antiS1IgG were noted for some measures (Appendix A). When correlations by ICU status and sex were examined, IFN-γ’s correlation with CCL7, TNF-β, TNF-α, and Fractalkine was lowest in ICU females (Figure 8b), whereas correlations in HCW males and females did not differ. IL-6 correlated significantly with chemokines CXCL9, CXCL10, CXCL1, and with cytokine IL-10 in female and male ICU patients, but not always with female non-ICU patients (Figure 8c). ncMono cell numbers were correlated with CXCL9 and IL-6 levels in ICU but not in non-ICU patients (Figure 8d), and pDC2 and DC2 cell numbers were also correlated in ICU but not non-ICU patients (Figure 8e).

## 4. Discussion

In our analysis of the data from the IMPACT cohort, we have several novel observations. We found that obesity is a significant risk factor. We delineated immune signatures unique to deceased patients, ICU, and non-ICU patients. We found that only 6/143 immune measures differed by sex in healthy controls. Of these six measures, four (CCL5, CXCL13, IL-6+CD4, and CD4Temra) were higher, and two measures (Eotaxin-3 and IL-21) were lower in healthy females versus males. Only 2/6 measures remained different in male and female COVID-19 patients; CD4Temra levels increased further in female COVID-19 patients versus HCW females, whereas CCL5 levels decreased in female and increased in male COVID-19 patients versus female and male HCW, respectively. De novo sex differences appeared in levels of twelve measures, while sex differences in four measures that were apparent in HCW disappeared after COVID-19 onset. Of the 14/143 measures that differed between the sexes, seven were greater in female COVID-19 patients including IFN-α2, CCL17, Fractalkine, CD4Temra, naïve CD8 T cells, T cells, and T cell numbers, whereas seven measures that were lower in female versus male patients included LIF, CCL5, EGF, Eotaxin, Eotaxin-3, IL-18, and cMono. We also found several measures that changed in the same direction in males and females, with DCs numbers being significantly lower after COVID-19 onset.

In the IMPACT cohort, some patients were classified in one or more of four different categories of risk factors for COVID-19. The original study reported that extreme BMI correlated with an increased relative risk of mortality [17], yet BMI was not considered as a risk factor and was instead adjusted for. Obese patients had the worst clinical score and 75% of SARS-CoV-2-negativepatients who were in ICU had a BMI > 30. The average BMI and clinical score for deceased patients who were all in ICU was 37 and 4.5, respectively, arguing that obesity/BMI should be classified as a major risk factor for health outcomes and should not be adjusted for, at least in COVID-19 patients. BMI, risk factor count, ICU status, clinical score, treatment count, days from corticosteroid end, and saliva load were variables that differed significantly between ICU and non-ICU patients and contributed to changed measures and health outcomes. IL-6 was the most changed measure across many variables, highly increased in deceased, ICU, and non-ICU patients, regardless of biological sex, and correlated most with CCL1. Viral load in saliva or nasopharyngeal swabs correlated with distinct cytokines/chemokines including IFN-γ, TNF-α, and CCL8 with nasopharyngeal load reaching 41% inverse correlation with AntiS1IgG in all patients. Curiously, some patients with multiple time points were negative for SARS-CoV-2 to start with but became positive while hospitalized and then were negative (0 value for viral load) again. Yet, others were positive to start with and became negative or had missing values. This suggests that either these patients were false positives and misclassified as COVID-19 patients or the tests were incorrect/inconclusive. Anti-S1IgG and IgM levels were barely over the limit of detection in several patients, and it is unclear whether the concentrations detected could be quantified reliably in the majority of the patients [17]. Even patients with high viral load did not necessarily have high anti-S1 titers. While it is stated that all patients were tested to be SARS-CoV-2-positive during the initial screen, those data are not provided, making it difficult to evaluate whether the patients were indeed truly ill with COVID-19. It is important to verify that the patients were indeed positive for SARS-CoV-2 as several of the patients had no viral load (missing or zero value; Appendix A) in their nasopharyngeal or saliva samples as determined from the raw data Table 41586_2020_2700_MOESM1_ESM provided by the authors. Given the seriousness of the situation at the time of these publications in mid 2020, it is surprising that the patients were not confirmed to be SARS-CoV-2^+^ before being included in the analysis as COVID-19 patients.

The nasopharyngeal viral load correlated negatively with AntiS1IgG suggesting the expected delay in appearance of viral-specific antibodies after days from infection. Absence of both viral load and AntiS1IgG/IgM in a number of patients supports the notion that they were SARS-CoV-2-negative and their inclusion may confound findings. Interestingly, saliva and NP loads correlated with distinct cytokines/chemokines suggesting location-specific activation of immune responses that help fight the invading pathogens. The saliva load correlated with CCL8, a chemokine that activates leukocytes and binds with high affinity to the receptor CCR5. CCL8 is known to be a potent inhibitor of HIV1 by competing for binding to CCR5 [27,28], which also serves as a co-receptor for HIV1, suggesting one mechanism to fight the invading virus. CCL8 was highly correlated with IL-10 and IFN-γ.

Lucas et al. [16] reported elevated levels of IFN-λ2, IFN-α2, IL-1RA, CCL1, MCSF, IL-2, IL-16, and CCL21 in deceased patients. We also found increased levels of all these cytokines except for IFN-λ2 in deceased patients, but all these cytokines were also elevated in non-ICU patients with moderate disease in our analysis. While all deceased patients were also ICU patients, in our analysis, patients who ultimately died of COVID-19 experienced significant changes in 11 immunological measures that were unique and not shared by other ICU or non-ICU COVID-19 patients. These measures included increased levels of IL-2, IL-1α, IL-17A, IL-1β, IL-12, IL-3, FGF2, Fractalkine, and IFN-γ^+^CD8^+^ T cell numbers and decreased numbers of IL-2^+^CD4^+^ and TNF-α^+^CD4^+^ T cells. Fractalkine was the only measure that also exhibited sex differences in expression levels with female COVID-19 patients exhibiting higher levels than male patients. Of note, female HCW also trended to have greater expression of Fractalkine than male HCW, but the differences were not statistically significant. 

Fractalkine (CX3CL1), a chemokine that alters the leukocyte adhesion mechanism to render their association with proteoglycans and other adhesion molecules irrelevant and modulates extravasation through the vascular wall, was highly correlated with IFNγ. Fractalkine, TNFα, SCF, and other cytokines were increased/decreased in patients with risks versus no risk and in female versus male with risks, except in cancer treatment and immunosuppressed patients. In bronchial epithelial cells, IFNγ stimulates Fractalkine expression, which in turn promotes adherence of blood mononuclear cells to the monolayers of bronchial epithelial cells [29]. Increases in Fractalkine levels may also contribute to lung pathology in COVID-19 patients. Dendric and NKT cell populations were highly decreased in patients versus HCW. IL-10 levels were lower in HCW female versus HCW male controls. IL-10 levels increased in the early and late phases of the disease; IL-10 has dual function, and its timing and spatiotemporal expression determines anti- or pro-inflammatory effects. These specific signatures when examined in depth could help understand immune mechanisms and “misfiring” that underlie differential outcomes between the sexes even at identical clinical scores taking risks and other variables into consideration but not adjusting for them.

Effector CD8^+^ T cells play a critical role in host defense against viruses and undergo clonal expansion after encounter with a pathogen. Naïve CD8^+^ T cells differentiate into lytic effector cells upon being stimulated a third time by cytokines such as IL-12 or type I interferons. Effector GzB^+^CD8^+^ T cells store perforin and granzyme B (GzB) in their cytosolic granules and release these lytic granules upon antigen encounter [30]. IL-18 was increased in males vs. females in all patients, and we replicated these findings; however, in ICU patients, IL-18 levels were higher in females and remained higher even after BMI adjustment. Taken together, our analysis shows that deceased patients had increased GzB^+^CD8^+^ T cells and concomitant increases in IFN-γ serving to activate the cytolytic function of GzB^+^CD8^+^ T cells.

ICU patients experienced significant changes in five measures that were not shared with deceased or non-ICU patients. TNF-β, CCL7, and FLT3L levels were increased whereas decreased measures included NKT cells and CD8 T cells. Three of these measures were changed in the opposite direction in male and female ICU patients, but the differences were not statistically significant, except for NKT cells. Non-ICU patients experienced significant changes in 10 measures that were unique and not different in deceased and ICU patients. Lucas et al. [16] reported higher levels of type 2 cytokines associated with eosinophilia, such as IL-5 and IL-13, in patients with severe disease but did not find significant differences in IL-4 levels. In agreement with their findings, we also found higher levels of IL-5 in non-ICU (moderate) and ICU (severe) patients, but IL-13 levels were only significantly increased in male ICU patients. It is possible that some changes in less abundant proteins like IL-4 may have been missed by Lucas et al. due to log transformation. In our analysis, IL-4 was significantly decreased in ICU patients when using the untransformed, left-skewed values but lost significance when using the optimal exponent 0.065 or log-transformed values.

The patients were given at least four different treatments for COVID-19, and some received multiple treatments, whereas others none. Potent immunomodulators such as Tocilizumab and corticosteroids were used, yet their effectiveness in reducing cytokine levels or modulating immune cell numbers in disease severity was not examined in any of the reports [16,17,31]. Our reanalysis suggests that none of the early treatments were effective in reducing levels of key proinflammatory cytokines such as IL-6, that are key components of the “cytokine storm”. More importantly, none of the treatments appear to reduce clinical symptoms or health outcomes. Such an analysis would have benefited the community and efforts could have been diverted and focused on other treatments.

IL-6 was the most significantly increased and changed cytokine across variables that correlated with clinical score and most increased in deceased male patients, but its levels in deceased female patients did not differ from those in ICU. Many chemokines and cytokines such as CCL1, CCL21, CXCL10, SCF, Fractalkine, and IL-10 correlated with each other in several groups under various biological variables such as DFSO, clinical score, treatments, and risks, suggesting that these immunological measures should be investigated in greater depth. In female vs. male patients’ comparison (Pt.♀/Pt.♂), only BMI and IL-16 were significantly different. IL-6 was most increased in deceased and ICU patients, but levels did not differ in female and male patients. IL-6 strongly correlated with IFN-λ2 (IFNL2orIL-28) and CCL1. SCF and IL-2 were most decreased in female CAC patients and correlated inversely with IFN-λ2 and FGF2. IFN-λ2’s role in immunoprotection is well recognized [32,33]. CD8Tem and CD8Tcm were the most decreased measures and they correlated strongest in deceased patients (r = 0.84, *p* = 10^−11.3^). CD8 T cells respond to cognate antigen and individuals in whom these memory CD8 T cells persist long-term and are often better protected against invading pathogens including viruses, bacteria, and protozoans [34].

Days 7–10 are a critical period for switching between recovery or going on to being critically ill [33]. We next determined how key variables such as DFSO, risk factors including obesity (BMI ≥ 30), treatments received, treatment counts, clinical score, ICU status, and outcomes “impacted” the significantly changed immunological signature identified in ICU female and male patients in confirmed SARS-CoV-2^+^ patients compared with HCW controls. In male CAC patients, CXCL10, a CXCR3 ligand, was the most significantly changed and CD8Tcm was most changed relative to non-CAC male patients. CXCL10 is involved in the generation of parasite-specific CD8 T-cell-mediated immune responses, and CXCL10 expression in the central nervous system regulates antibody-secreting cell accumulation during SARS-CoV-2-induced encephalomyelitis [35]. Elevated CXCL10 levels are correlated with COVID-19-related ARDS and neurological complications and is considered a predictive biomarker of COVID-19 severity and disease progression. SCF, an essential hematopoietic cytokine, interacts with other cytokines to preserve the viability of hematopoietic stem and progenitor cells. SCF was markedly decreased in female CAC patients along with IL-2, IL-23, IL-16, VEGFA, macrophages, T cells, NKT cells, CD8Tem, and several others, whereas total and ncMono, IL6+CD4, and TNF-α were increased.

This study has some limitations: we did not have access to all data, including clinical measures that could improve prediction capabilities or confirm whether other markers such as ferritin were changed in this cohort and correlated with outcomes. Sample size is small to be analyzed for variables such as coagulopathy and DFSO. Of note, not all measures were detected or quantifiable in all patients. For examples, IL-2 was detected in plasma of only ~12% of female and ~26% of male patients, and IFN-γ in ~61% of female and ~78% of male patients. Imputed values for missing data can mean that either the patients did not have those measures, or the assay was not sensitive enough. If former, imputed values can be misleading as values under the limit of detection and/or quantification suggests that those cytokines/chemokines were only secreted by a subset of patients depending upon their comorbidities and/or other health status. Decreased numbers of immune cells in the blood could also be due to increased infiltration/migration of specific immune cells by tissues such as the lungs or secondary lymphoid organs. A number of measures, despite being shared between the sexes, did not necessarily correlate to the same degree in female and male ICU patients. For example, CCL21 correlated with CCL1, SCF, and Fractalkine to a much lesser degree in female as compared with male ICU patients, suggesting nuanced regulation and signaling that can be easily missed in sex aggregated analysis.

## 5. Conclusions

In conclusion, we report several novel findings that were missed in the original articles: first, the immune signature of ICU and deceased patients is strikingly different than that of non-ICU patients, with notable absence of differences in many usual suspects such as IL-6, IFN-α2, CCL1, and CCL2 between non-ICU and ICU patients, whereas CCL5 exhibited sex differences in its expression levels in both HCW and COVID-19 patients. Second, none of the treatments, including immunomodulators such as Solu-medrol (corticosteroid) and Tocilizumab, decreased levels of IL-6 or key cytokines/chemokines implicated in cytokine storm, nor did Remdesivir or hydroxychloroquine. Third, dendritic (cDC1s, cDC2s, and pDCs) and NKT cells were decreased in all COVID-19 patients regardless of sex, a finding confirmed later [36]. Fourth, men and women shared many measures that did not differ with sex as a variable but were influenced differentially by variables such as risk factors, clinical score, and treatments. Sex differences in at least nine immunological measures were also evident in healthy controls. Fifth, patients with obesity as a risk factor had the most changes in all measures and worst outcomes, including mortality, whereas patients who were immunosuppressed experienced the greatest changes in immunological signatures including effector memory T cells- CD4Temra involved in protective immunity. Taken together, our multi-dimensional integrated analyses revealed many significant findings that were missed in the original articles. We provide support that sex-aggregated analysis, which has been the norm for clinical studies, is often misleading. Most animal studies in the past predominantly used one sex (male) and, hence, the data were not confounded, but with changes in NIH policy with regards to sex as a biological variable (SABV) [37], when using both sexes, researchers often perform combined analysis, thereby missing key findings. Similarities and differences should both be reported and are essential for understanding divergent pathways that lead to similar health outcomes.

## Figures and Tables

**Figure 1 cells-12-02591-f001:**
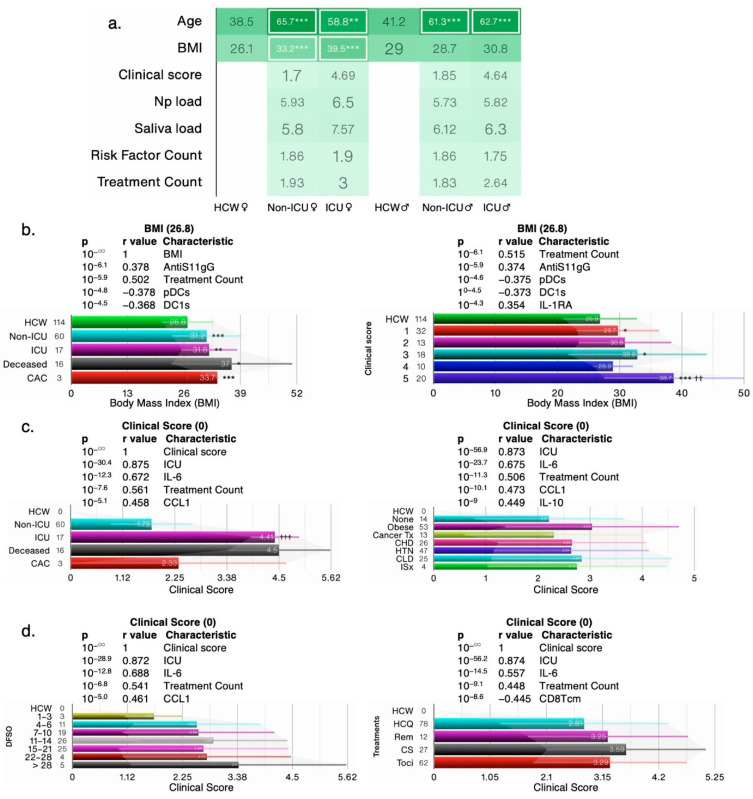
Obesity is a risk factor for COVID-19; (**a**) The average clinical score in test groups based on intensive care unit (ICU) admission/outcome and COVID risk factors segregated by sex. (**b**–**d**) Bar charts showing the average BMI and clinical scores in test groups based on ICU admission/outcome, days from COVID-19 symptom onset (DFSO), treatment, and risk factors. Cancer treatment (Tx) received in the prior year; CHD: chronic heart disease; HTN: hypertension; CLD: chronic lung disease; ISx: immunosuppressed patients. HCW: healthcare workers healthy controls; HCQ: hydroxychloroquine; Rem: Remdesivir; CS: high dose of corticosteroid; Toci: Tocilizumab; CAC: COVID-19-associated coagulopathy. The N/group in the bar charts (**b**–**d**) is shown on the *x*-axis. Error bars: ± standard deviation (SD). ***, **, and * denote *p* < 0.001, 0.01, and 0.05 versus control, respectively, whereas ^†††^ and ^††^ denote *p* < 0.001 and 0.01 versus the preceding test group (one bar above), using Welch’s *t*-test.

**Figure 2 cells-12-02591-f002:**
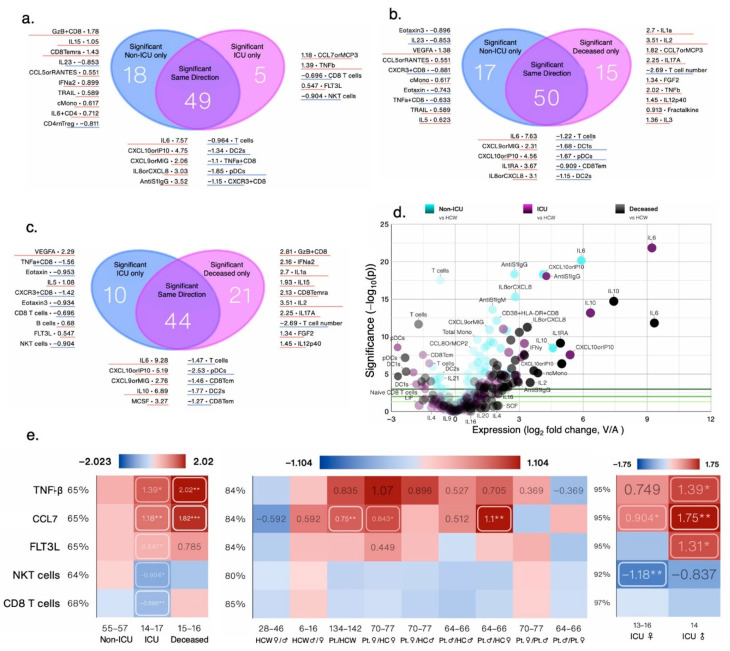
Venn diagrams contrasting significantly changed measures between non-ICU and ICU patients (**a**), non-ICU and deceased patients (**b**), and ICU and deceased patients (**c**), each compared to healthy controls (HCW), and listing the most changed symbols for each Venn diagram segment, sorted by impact score (average rank across *p*-value, relative change, and absolute change; not shown), with log_2_-fold change versus non-ICU patients of the same sex shown. (**d**) Composite volcano plot showing changes in each group versus healthy controls (HCW). (**e**) Heat map showing all significantly changed measures in ICU patients that were not shared with either non-ICU or deceased patients by patient status and sex. Numbers below the cells denote the number of data points in that group (n). Values shown are log_2_-fold change versus the respective control group (red: increased; blue: decreased), and value labels were drawn for values greater than 33% of the heat map’s maximum value. The maximum value does not include outliers (highlighted in yellow) defined as less than Q1 − 1.5(IQR) or greater than Q3 + 1.5(IQR). Labels for values less than 16.67% of the maximum are drawn in black for legibility. *, **, and *** denote *p* < 0.05, 0.01, and 0.001 per Welch’s *t*-test.

**Figure 3 cells-12-02591-f003:**
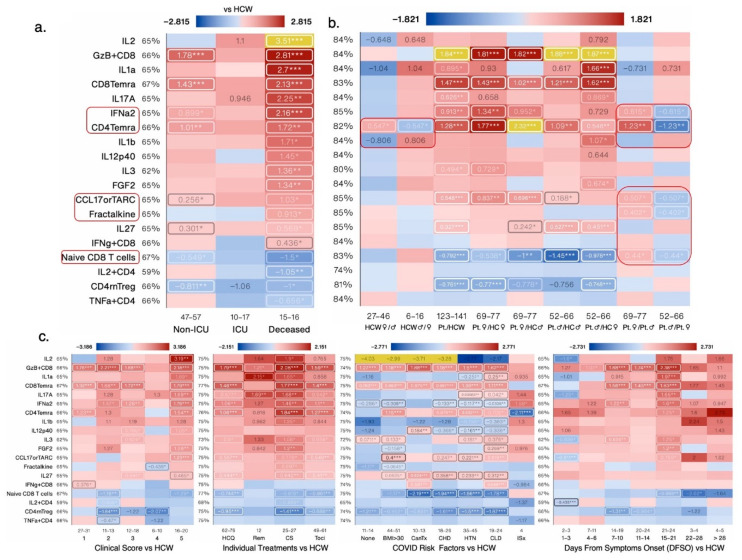
Heat map showing all measures from Figure 2b,c that were significantly changed in deceased patients, in groups based on the following: (**a**) ICU admission/outcome; (**b**) Patient status and sex; (**c**) Clinical score, treatments, risk factors and days from symptom onset (DFSO). All values shown are average log_2_-fold change (untransformed) versus the respective control group average, outlier-adjusted. Numbers below the cells denote the number of data points in that group (n). Values shown are log_2_-fold change versus the respective control group (red: increased; blue: decreased), and value labels were drawn for values greater than 33% of the heat map’s maximum value. The maximum value does not include outliers (highlighted in yellow) defined as less than Q1 − 1.5(IQR) or greater than Q3 + 1.5(IQR). Labels for values less than 16.67% of the maximum are drawn in black for legibility. Cancer treatment (Tx) received in prior 1 year; CHD: chronic heart diseases; HTN: hypertension; CLD: chronic lung diseases; ISx: immunosuppressed patients. HQ: hydroxychloroquine; Remdes: Remdesivir; Cort: high dose of corticosteroid; Toci: Tocilizumab. V/A: value-to-average. IQR: interquartile range. (**d**) Bar charts showing the average GzB^+^CD8 cell numbers in non-ICU, ICU, and deceased patients compared with HCW; significant changes were seen in non-ICU and deceased patients. Scatter plots of the top 4 measures that correlated with GzB^+^CD8 cell numbers in all patients, each showing the goodness-of-fit (R^2^), Pearson’s r, and *p*-values for all test groups. *, **, and *** denote *p* < 0.05, 0.01, and 0.001 per Welch’s *t*-test. Measures exhibiting sex differences are highlighted within boxes.

**Figure 4 cells-12-02591-f004:**
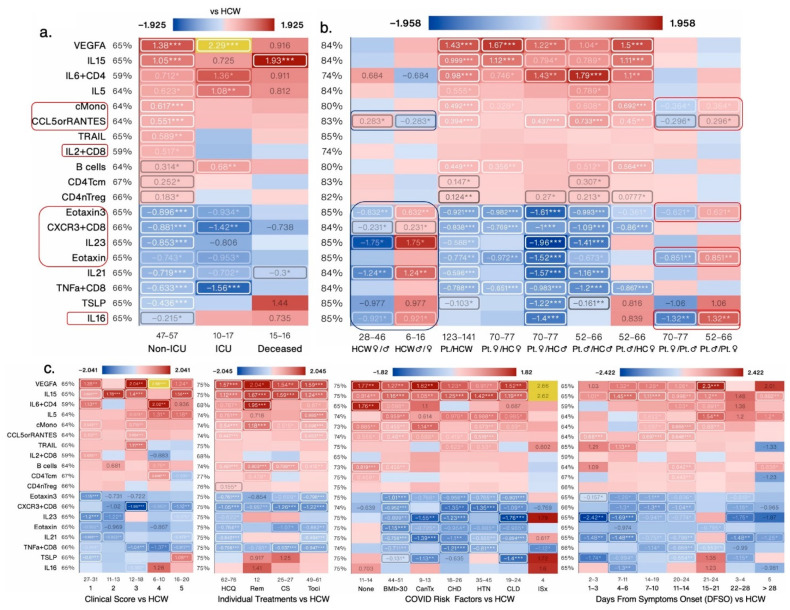
Heat map showing all measures from Figure 2a,b that were significantly changed in non-ICU patients, in groups based on the following: (**a**) ICU admission/outcome’ (**b**) Patient status and sex; (**c**) Clinical score, treatments, risk factors, and days from symptom onset (DFSO). All values shown are average log_2_-fold change (untransformed) versus the respective control group average, outlier-adjusted. Numbers below the cells denote the number of data points in that group (n). Values shown are log_2_-fold change versus the respective control group (red: increased; blue: decreased), and value labels were drawn for values greater than 33% of the heat map’s maximum value. The maximum value does not include outliers (highlighted in yellow) defined as less than Q1 − 1.5(IQR) or greater than Q3 + 1.5(IQR). Labels for values less than 16.67% of the maximum are drawn in black for legibility. *, **, and *** denote *p* < 0.05, 0.01, and 0.001 per Welch’s *t*-test. HCW: healthcare worker; Pt: COVID-19 patients; Cancer treatment (Tx) received in prior 1 year; CHD: chronic heart diseases; HTN: hypertension; CLD: chronic lung diseases; ISx: immunosuppressed patients. HQ: hydroxychloroquine; Remdes: Remdesivir; Cort: high dose of corticosteroid; Toci: Tocilizumab. V/A: adjusted value-to-average. IQR: interquartile range. Measures exhibiting sex differences are highlighted within boxes.

**Figure 5 cells-12-02591-f005:**
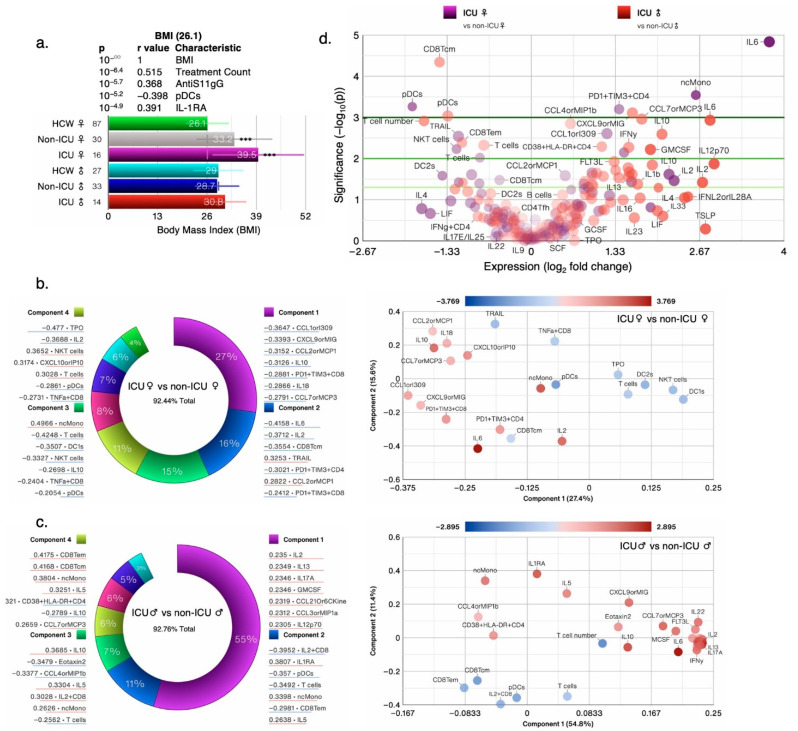
Nuanced immunological profile of male and female COVID-19 ICU patients: (**a**) Bar chart showing the average BMI in test groups based on ICU admission and sex. (**b**) PCA donut plots showing the primary components necessary to explain at least 90% of the variance in female (**b**) and male (**c**) ICU patients, and 7 symbols most correlated with each of the first four primary components (left),and PCA biplots for the first two components (right), with the color of points denoting log_2_-fold change versus non-ICU patients of the same sex. (**d**) Volcano plots comparing female (purple circles) and male (red circles) ICU patients to non-ICU patients of the same sex. Error bars: ± standard deviation (SD). *** denotes *p* < 0.001.

**Figure 6 cells-12-02591-f006:**
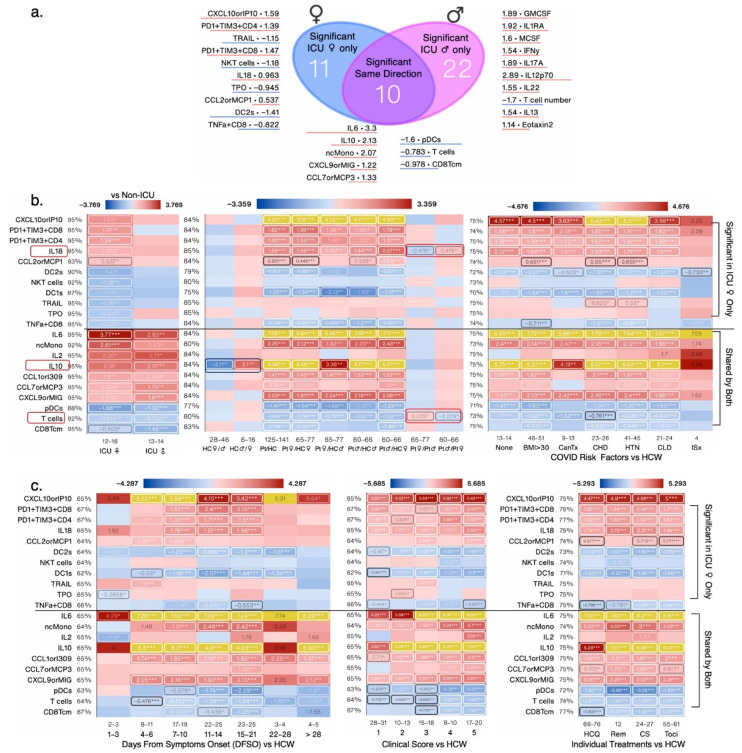
Impact of various biological variables on significantly changed measures in female ICU patients: (**a**) Venn diagram contrasting significantly changed measures between female (left) and male (right) ICU patients each compared to non-ICU patients and listing the 10 most changed symbols for each Venn diagram segment, sorted by impact score (average rank across *p*-value, relative change, and absolute change; not shown), with log_2_-fold change versus non-ICU patients of the same sex shown. (**b**) Heat map showing all 11 measures from Figure 6a that were significantly changed in ICU females only and all 10 shared measures in ICU females and males compared to non-ICU females and males, respectively, by patient status, sex, and COVID-19 risk factors, and by DFSO, clinical score, and treatment (c). Numbers below each cell denote the number of data points in that group (n). All values shows are average log_2_-fold change (untransformed) versus the respective control group average, outlier-adjusted. Value labels were drawn for values greater than 33% of the heat map’s maximum value. The maximum value does not include outliers (highlighted in yellow) defined as less than Q1 − 1.5(IQR) or greater than Q3 + 1.5(IQR). Labels for values less than 16.67% of the maximum are drawn in black for legibility. *, **, and *** denote *p* < 0.05, 0.01, and 0.001 per Welch’s *t*-test. HC: healthcare workers as healthy controls; Pt: all COVID-19 patients; Cancer treatment (Tx) received in prior 1 year; CHD: chronic heart diseases; HTN: hypertension; CLD: chronic lung diseases; ISx: immunosuppressed patients. HQ: hydroxychloroquine; Remdes: Remdesivir; Cort: high dose of corticosteroid; Toci: Tocilizumab. V/A: value-to-average. IQR: interquartile range. Measures exhibiting sex differences are highlighted within boxes.

**Figure 7 cells-12-02591-f007:**
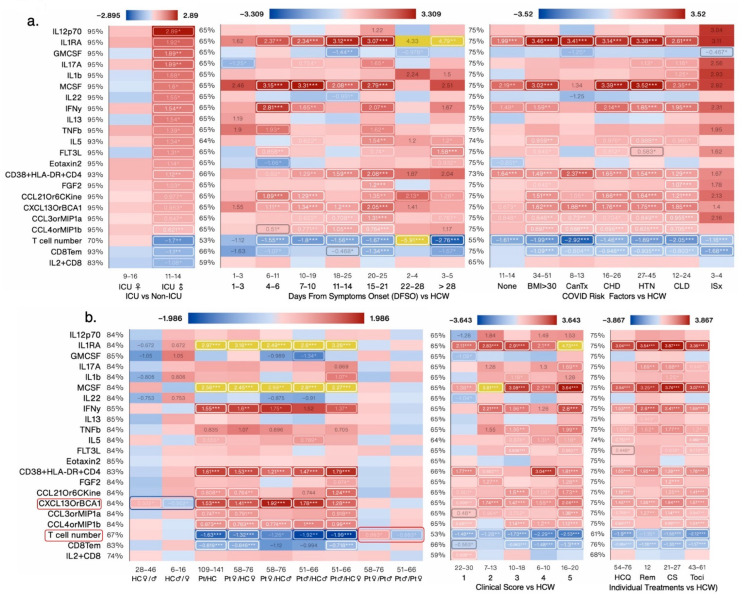
Impact of various biological variables on significantly changed measures in male ICU patients. (**a**) Heat map showing all 22 measures from Figure 6a that were significantly changed in ICU males only compared to non-ICU males, by patient status, sex, and COVID-19 risk factors, and DFSO, clinical score, and treatment (**b**). Numbers below each cell denote the number of data points in that group (n). All values shows are average log_2_-fold change (untransformed) versus the respective control group average, outlier-adjusted. Value labels were drawn for values greater than 33% of the heat map’s maximum value. The maximum value does not include outliers (highlighted in yellow) defined as less than Q1 − 1.5(IQR) or greater than Q3 + 1.5(IQR). Labels for values less than 16.67% of the maximum are drawn in black for legibility. *, **, and *** denote *p* < 0.05, 0.01, and 0.001 per Welch’s *t*-test. HC: healthcare workers as healthy controls; Pt: all COVID-19 patients; Cancer treatment (Tx) received in prior 1 year; CHD: chronic heart diseases; HTN: hypertension; CLD: chronic lung diseases; ISx: immunosuppressed patients. HQ: hydroxychloroquine; Remdes: Remdesivir; Cort: high dose of corticosteroid; Toci: Tocilizumab. V/A: value-to-average. IQR: interquartile range. Measures exhibiting sex differences are highlighted within boxes.

**Figure 8 cells-12-02591-f008:**
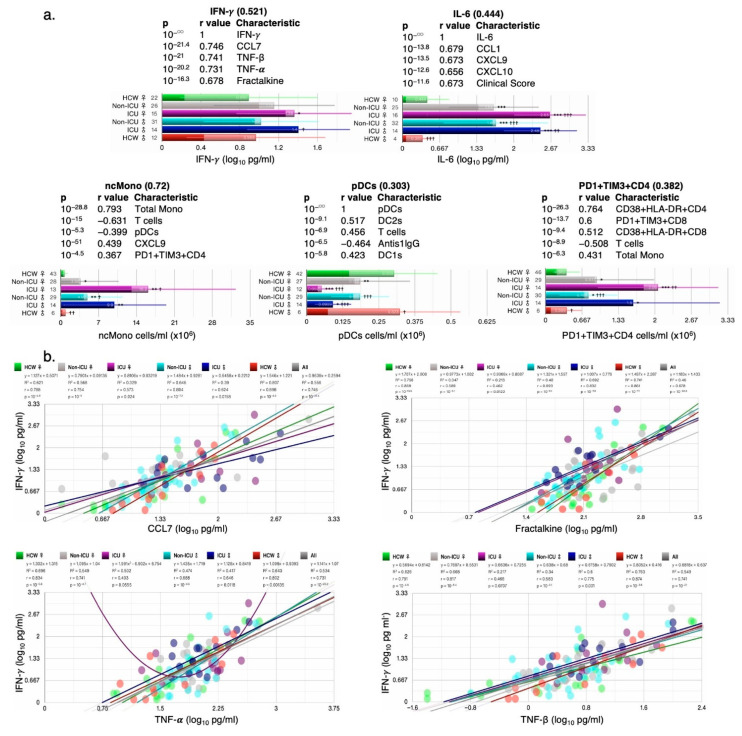
Sex differences in correlation with key cytokines/chemokines. (**a**) Bar charts showing the average abundance of interferon gamma (IFN-γ), interleukin-6 (IL-6), nonclassical monocytes (ncMono), plasmacytoid dendritic cells (pDCs), and CD4^+^ T cells positive for programmed cell death protein 1 (PD-1) and hepatitis A virus cellular receptor 2 (TIM3) (PD1^+^TIM3^+^CD4^+^). Scatter plots for the four measures most significantly correlated with IFN-γ (**b**), IL-6 (**c**), ncMono (**d**), and pDCs (**e**), each showing the goodness-of-fit (R^2^), Pearson’s r, and *p*-values for all test groups. Numbers on *x*-axis in bar charts in (**a**) denote the actual number of patients in which the measures were detected. Not all measures were detected in all patients. Total N/group: HCW♀: 87; non-ICU♀: 30; ICU♀: 16; non-ICU♂: 33; ICU♂: 14; HCW♂: 27. Error bars: ± standard deviation (SD). ***, **, and * denote *p* < 0.001, 0.01, and 0.05 versus control, respectively, whereas ^†††^, ^††^, and ^†^ denote *p* < 0.001, 0.01, and 0.05 versus the preceding test group (one bar above), using Welch’s *t*-test.

## Data Availability

All data are contained within the manuscripts or supplemental data. Original raw data was accessed by the authors on 23 May 2023 and can be downloaded at: https://www.ncbi.nlm.nih.gov/pmc/articles/PMC7477538/#SD2.

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
