# Peer review of "Immunological Misfiring and Sex Differences/Similarities in Early COVID-19 Studies: Missed Opportunities of Making a Real IMPACT"

_cells, 2023, doi:10.3390/cells12222591_

Round 1

Reviewer 1 Report (Previous Reviewer 1)

Comments and Suggestions for Authors

1. CCL8orMCP2 should be corrected to CCL8. IL-10 instead of IL10 and IL-6 instead of IL6, similarly for each other interleukin.
2. Figure 6C - more variables are declared on the left than rows in the figure.
3. How is HIV1 relevant to the study? (line 406, disucssion)
4. Discussion is chaotic e.g. 1st paragraph ends with discussion on SARS-CoV2 +/- status, 2nd paragraph starts with CCL8 and then suddenly goes back to +/- status.
5. Line 443 should read CD8Tcm
6. Iines 485-490. This is a well-known problem with some cytokines in serum. Undetectable level of IL-2 does not necessarily mean that this cytokine was not produced or was significantly less abundant in said patients. IL-2 is one of the cytokines crucial for T cell survival, thus it is usually quickly bound by IL-2 receptor on the surface of T cells and may be hard to quantify in serum.
7. "Decreased numbers of immune cells in plasma" - by definition plasma is cell-free!
8. "increased uptake of specific immune cells by tissues" - we normally talk about migration to tissues, this is an active process that involves the action of immune cells.
9. The single most crucial limitation of the study is the construction of the study per se. It is merely a sophisticated statistical analysis that proposes some correlations but does not give us any real biological insight into either disease or treatment.

Comments on the Quality of English Language

No major issues

Author Response

  1. CCL8orMCP2 should be corrected to CCL8. IL-10 instead of IL10 and IL-6 instead of IL6, similarly for each other interleukin. 

Response. We used the captions reported in the original dataset and as reported in the two original Nature publication by Lucas and Takahashi et al. To be consistent with the original report, we kept the original symbol format. Moreover, it would be a lot of work to change all the symbol captions in the raw data file and if someone else wanted to cross- check, we thus used the symbols provided in the raw data files. However, in our manuscript and in the legends, we have changed the symbols as suggested.

  1. Figure 6C - more variables are declared on the left than rows in the figure.

Response. We apologize for this oversight. The figure has been revised. We found that many of the changed measures overlapped between males and females, so we eliminated several duplicates and reduced the number of panels in Fig. 6. We also noted that we inadvertently included SARS-CoV-2 negative patients in the non-ICU cohort, so while revising figure 6, we excluded CoV2 negative patients from all analyses in this study.

  1. How is HIV1 relevant to the study? (line 406, disucssion)

Response. SARS-CoV-2 is an RNA virus and appears to contain signatures that are very similar to HIV’s gp120. Both also share many shared immunological features. Moreover, discussion is appropriate to speculate about potential mechanisms.

  1. Discussion is chaotic e.g. 1st paragraph ends with discussion on SARS-CoV2 +/- status, 2nd paragraph starts with CCL8 and then suddenly goes back to +/- status. 

Response. We thank the reviewer for this feedback. Certainly, these are stylist issues. However, the point is well taken and we have extensively revised our discussion as well as the entire manuscript.

  1. Line 443 should read CD8Tcm

Response. Thank you for finding this typo. This has been corrected.

  1. Iines 485-490. This is a well-known problem with some cytokines in serum. Undetectable level of IL-2 does not necessarily mean that this cytokine was not produced or was significantly less abundant in said patients. IL-2 is one of the cytokines crucial for T cell survival, thus it is usually quickly bound by IL-2 receptor on the surface of T cells and may be hard to quantify in serum. 

Response. We agree with the reviewer and that is why we think that missing values shouldn’t be excluded or imputed. We were trying to make this point and have incorporated the example provided in our discussion to make this point clearer (lines 526-532).

  1. "Decreased numbers of immune cells in plasma" - by definition plasma is cell-free!

Response. We apologize for the oversight. This has been corrected.

  1. "increased uptake of specific immune cells by tissues" - we normally talk about migration to tissues, this is an active process that involves the action of immune cells.

Response. Writing styles and word usage often differ and we take the point that tissue migration or infiltration are better terminology than uptake by tissues. We have accordingly revised our manuscript.

  1. The single most crucial limitation of the study is the construction of the study per se. It is merely a sophisticated statistical analysis that proposes some correlations but does not give us any real biological insight into either disease or treatment.

Response. We thank the reviewer for finding our analysis sophisticated. Surely, for a complex dataset like this, a complex and integrated analysis is required to understand the significance of the dataset. There are many critical findings that the original report missed, and we provide insights about risk factors and early treatments as well as shed light on sex differences and similarities that exist in men and women- both healthy and those with COVID-19. Our analysis that segregates non-ICU and ICU patients and Deceased ICU patients is critical in understanding immunological measures that could be critical in determining one’s fate- recovery versus death. The reviewer is correct that a full picture of biological insights cannot be attained with just the immunological aspect.

Reviewer 2 Report (Previous Reviewer 2)

Comments and Suggestions for Authors

In order to say that any parameter is significantly different between one group and another group, it is more accurate to use parametric or non-parametric statistical methods. Correlation analyzes only indicate the level of relationship between variables. However, I think that it is necessary to perform a regression analysis and calculate the odd-ratio values of the variables in order to determine the effect of the variables on the severity of COVID-19.

Author Response

In order to say that any parameter is significantly different between one group and another group, it is more accurate to use parametric or non-parametric statistical methods. Correlation analyzes only indicate the level of relationship between variables. However, I think that it is necessary to perform a regression analysis and calculate the odd-ratio values of the variables in order to determine the effect of the variables on the severity of COVID-19.

Response. We used the non-parametric significance test Welch’s t-test for the following reasons: a) variable group sizes (³different variances between groups), and b) potentially unequal variances between males and females even with equal group sizes. Univariate linear or polynomial regression was performed for symbol pairs as described in Methods; the resulting regression curves and their equations are shown in Figures. Calculating odds-ratios for COVID-19 severity was beyond the scope of our work; our aim was to point out notable/important correlations between symbols that were missed in the original study.

Reviewer 3 Report (New Reviewer)

Comments and Suggestions for Authors

Thank you for the opportunity to review the manuscript “Immunological misfiring and sex differences/similarities in early COVID-19 studies: missed opportunities of making a real IMPACT  (cells-2638127).

Authors analysed publicly available data from the Yale IMPACT cohort to address immunological misfiring and sex differences in early COVID-19 ICU patients by taking various biological variables into account. They considered biological variables such as days from symptoms onset, pre-existing risk factors, including obesity, and early COVID-19 treatments that will influence the immunological measures.

The work is important, but some parts of the paper must be carefully revised.

One of my concerns related to the analysis continues to be related to the introduction literature. Greater details about the mentioned previous studies results are needed than currently provided that builds the case for having conducted the current study. This could also strengthen the discussion, as it is quite common to refer to findings from those studies relative to the current study findings in the discussion and conclusions sections.

Authors mentioned aims but no clear research question, no hypothesis at the end of introduction.

The introduction should also be written in a way that is easier for the reader to understand. A table with all the abbreviations would also be very helpful. Figure 6a is very difficult to read.

Discussion:  It would also be helpful to explain how the current study differs from previous studies in more detail. This section should be streamlined to focus specifically on key findings in discuss them.

In the conclusion, the five new aspects that were gained are only listed.

What is the implication for further studies?

Comments on the Quality of English Language

Language: minor editing is required

Author Response

Thank you for the opportunity to review the manuscript “Immunological misfiring and sex differences/similarities in early COVID-19 studies: missed opportunities of making a real IMPACT  (cells-2638127).

Authors analysed publicly available data from the Yale IMPACT cohort to address immunological misfiring and sex differences in early COVID-19 ICU patients by taking various biological variables into account. They considered biological variables such as days from symptoms onset, pre-existing risk factors, including obesity, and early COVID-19 treatments that will influence the immunological measures.

The work is important, but some parts of the paper must be carefully revised.

One of my concerns related to the analysis continues to be related to the introduction literature. Greater details about the mentioned previous studies results are needed than currently provided that builds the case for having conducted the current study. This could also strengthen the discussion, as it is quite common to refer to findings from those studies relative to the current study findings in the discussion and conclusions sections.

Authors mentioned aims but no clear research question, no hypothesis at the end of introduction. The introduction should also be written in a way that is easier for the reader to understand. 

A table with all the abbreviations would also be very helpful. Figure 6a is very difficult to read.

Response. We appreciate this comment. As asked, we have now described key aspects of the previous study in the introduction and discussion. We state our in the introduction that there were so many biological variables that were ignored in the original study and inclusion of nearly 50% COVID-19 negative patients likely confounds many of the original findings.

We have defined all non-standard abbreviations used. Standard accepted abbreviations in the field of immunology such as those for cytokines/chemokines and T/B cell subsets and markers are not defined.

In order to provide bigger figure size, in this revised manuscript, we attach figures at the end of the document, rather than in the Results section.

Discussion:  It would also be helpful to explain how the current study differs from previous studies in more detail. This section should be streamlined to focus specifically on key findings in discuss them. In the conclusion, the five new aspects that were gained are only listed. What is the implication for further studies?

Response. We appreciate this comment. We have integrated our discussion and conclusion sections and discussed implications.

Round 2

Reviewer 1 Report (Previous Reviewer 1)

Comments and Suggestions for Authors

Authors have sufficiently addressed my questions.

Comments on the Quality of English Language

Acceptable

Reviewer 3 Report (New Reviewer)

Comments and Suggestions for Authors   The reviewer's relevant suggestions for improving the paper were made. Comments on the Quality of English Language

 Moderate editing of English language required.

This manuscript is a resubmission of an earlier submission. The following is a list of the peer review reports and author responses from that submission.

Round 1

Reviewer 1 Report

Comments and Suggestions for Authors

Authors present a reanalysis of already published dataset. Unfortuntaly, the majority of the study makes little sense as it is pure data mining with some random correlations as shown in figure 7. Some are obvious like total mono vs non-classical monocytes while others are hard to justify like fractalkine vs IFN (what IFN??). Authors lack basic understanding of immunology, they use some captions from the dataset e.g. CCL8orMCP2 instead of writing either CCL8 or MCP2 as those are two synonyms for the same protein!

1. I would suggest providing bigger figures. Currently, they are hard to read e.g. clusters in PCA analysis in fig. 3.
2. Lines 216-225. Authors should use some more descriptive approach for immunological part e.g. ncMono should be assigned their full name - non-classical monocytes.
3. Multiple typos can be noted
4. Figure 7 - what IFN? Why is this figure divided in such a way?
5. Lines 399-401 - this is the very definition of acquired immunity - every immunocompetent individual develops long-term memory T cells, both CD8 and CD4+.

Comments on the Quality of English Language

Significant corrections required. Article generally poorly written. Multiple typos. Poor flow.

Author Response

Comments and Suggestions for Authors

Authors present a reanalysis of already published dataset.

Response: There are many articles that use already published dataset for reanalysis to report findings that were not covered or missed by original studies.

Unfortuntaly, the majority of the study makes little sense as it is pure data mining with some random correlations as shown in figure 7. Some are obvious like total mono vs non-classical monocytes while others are hard to justify like fractalkine vs IFN (what IFN??).

Response. We respectfully disagree that the study makes little sense. We report several novel findings that were missed as summarized in the conclusion section. Correlations shown in Fig. 7 are far from random. Interferon gamma (IFNg- for some reason the Greek symbols didn’t show up in Fig. 7, but were specified in figure legend), IL6, ncMono, and pDC are all key immunological measures in COVID-19; dysregulated IL6 and IFNg levels are what has been proposed in the pathology of SARS-CoV-2 by many studies. Figure 7 shows top 4 correlations for each of those measures with other immunological measures present in the IMPACT dataset that we found in our analysis and that were missed. We are not sure how the reviewer can claim that “total mono and ncMono is obvious” as there is no correlation between total Mono and ncMono in healthy controls. Total Mono do not correlate with fractalkine but IFNg does. Furthermore, IFNg  is known to stimulate fractalkine expression (https://doi.org/10.1165/ajrcmb.25.2.4275). It is what the data shows and sometimes we might not have an explanation for an observation, but that doesn’t mean it is not valid or important. We provided explanation as to why we think correlation with fractalkine makes sense based on previous published studies (previous lines 423-428). The revised text now reads “In bronchial epithelial cells, IFNg stimulates fractalkine expression, which in turn promotes adherence of blood mononuclear cells to the monolayers of bronchial epithelial cells [24]. Increase in fractalkine levels may also contribute to lung pathology in COVID-19 patients”. We are unclear as to what exactly is the objection being raised by the reviewer for this correlation.

Authors lack basic understanding of immunology, they use some captions from the dataset e.g. CCL8orMCP2 instead of writing either CCL8 or MCP2 as those are two synonyms for the same protein!

Response. We respectfully disagree with the reviewer’s assessment. We used the captions reported in the original dataset and as reported in the original Nature publication. To be consistent with the original report, we kept the original format. Moreover, it would be a lot of work to change all the symbol captions in the raw data file and if someone else wanted to cross-check our findings, it would be difficult with changed captions.

  1. I would suggest providing bigger figures. Currently, they are hard to read e.g. clusters in PCA analysis in fig. 3. 

Response. We provided separate pdf figures. It is hard to make captions bigger in the embedded figure itself due to space limitations.

  1. Lines 216-225. Authors should use some more descriptive approach for immunological part e.g. ncMono should be assigned their full name - non-classical monocytes. 

Response. We thank the reviewer for pointing this out. We have defined abbreviations when first used; however, standard accepted abbreviations in the field of immunology are not defined.

  1. Multiple typos can be noted

Response. We apologize for the typos and have corrected a few that we found. It would have been helpful if the reviewer would have pointed out the typos.

  1. Figure 7 - what IFN? Why is this figure divided in such a way?

Response. It is IFNg (interferon gamma). For some reason, the Greek alphabets didn’t show up in Fig. 7 alone in the converted pdf. We have shown the top 4 correlations and we showed the two TNFs side-by-side. We are not clear what exactly is the reviewer objecting to.

  1. Lines 399-401 - this is the very definition of acquired immunity - every immunocompetent individual develops long-term memory T cells, both CD8 and CD4+. 

Response. We agree and again are not clear what exactly is the reviewer objecting to.

Comments on the Quality of English Language

Significant corrections required. Article generally poorly written. Multiple typos. Poor flow. 

Response. It would have been helpful if the reviewer would have pointed out a few typos and provided some feedback as to what the problem with the flow was. We have revised our manuscript based on the feedback provided by all the reviewers.

Reviewer 2 Report

Comments and Suggestions for Authors

Abstract

Avoid comments in this section. Mention the main purpose and main findings of the study. Finish by reporting only the most important results.

Introduction Section

- In this section, the basic infrastructure suitable for the hypothesis of the study should be given and the hypothesis of the study and which deficiency in the literature should be addressed. Here it should be mentioned which parameters/biomarkers will be compared. The role of these parameters in COVID-19 should be mentioned. Apart from this, methods, comments and discussions should not be entered in this section. Be sure to mention the main purpose and hypothesis of the study in this section. The introduction should be revised.

- I suggest you put the heading “Biological variables in the IMPACT cohort” in the “material” section. Such a presentation in the introduction is not appropriate.

There is important information in the following articles that point to the role and importance of laboratory markers in the diagnosis, prognosis and mortality of COVID-19. I suggest citing the key findings of these articles.

-Detection of Risk Predictors of COVID-19 Mortality with Classifier Machine Learning Models Operated with Routine Laboratory Biomarkers. Appl. Sci. 2022, 12, 12180. https://doi.org/10.3390/app122312180.

-Machine Learning Sensors for Diagnosis of COVID-19 Disease Using Routine Blood Values for Internet of Things Application. Sensors 2022, 22, 7886. https://doi.org/10.3390/s22207886.

-Effect of ferritin, INR, and D-dimer immunological parameters levels as predictors of COVID-19 mortality: A strong prediction with the decision trees. Heliyon, e14015. https://doi.org/10.1016/j.heliyon.2023.e14015.

Material and Method Section

The parameters that were compared between the patient groups in this study should be mentioned here. In addition, which analyzes are used for statistical comparisons between groups and how the distributions are controlled should be written in the statistical analysis section. In the statistical analysis section, it should be stated what purpose the methods were used for (for example, what was the principal component analysis used for). Here, only Pearson correlation analysis was used for correlation analysis. It may be in places where semen should be used. Did all variables satisfy the assumption of normality? Which statistical package program was used? information should be given about them.

Result Section

What exactly does Figure 5 and Figure 6 show? What do the numbers in the figures show? I'm having a really hard time figuring out what the methods are used for. Did you interpret the correlation results as an effect here? Correlation analyzes provide information about the presence and amount of positive/negative relationship between variables. However, I think that it is necessary to perform a regression analysis and calculate the odd-ratio values of the variables in order to determine the effect of the variables on the severity of COVID-19.

Discussion and Conclusions section

The limitations of the study should be mentioned. In the conclusion part, information about the most important findings should be given without entering the comments.

Comments on the Quality of English Language

Minor editing of English language required

Author Response

Comments and Suggestions for Authors

Abstract

Avoid comments in this section. Mention the main purpose and main findings of the study. Finish by reporting only the most important results.

Response. We thank the reviewer for this helpful critique and have revised the abstract.

Introduction Section

- In this section, the basic infrastructure suitable for the hypothesis of the study should be given and the hypothesis of the study and which deficiency in the literature should be addressed. Here it should be mentioned which parameters/biomarkers will be compared. The role of these parameters in COVID-19 should be mentioned. Apart from this, methods, comments and discussions should not be entered in this section. Be sure to mention the main purpose and hypothesis of the study in this section. The introduction should be revised.

- I suggest you put the heading “Biological variables in the IMPACT cohort” in the “material” section. Such a presentation in the introduction is not appropriate.

There is important information in the following articles that point to the role and importance of laboratory markers in the diagnosis, prognosis and mortality of COVID-19. I suggest citing the key findings of these articles.

-Detection of Risk Predictors of COVID-19 Mortality with Classifier Machine Learning Models Operated with Routine Laboratory Biomarkers. Appl. Sci. 2022, 12, 12180. https://doi.org/10.3390/app122312180.

-Machine Learning Sensors for Diagnosis of COVID-19 Disease Using Routine Blood Values for Internet of Things Application. Sensors 2022, 22, 7886. https://doi.org/10.3390/s22207886. 

-Effect of ferritin, INR, and D-dimer immunological parameters levels as predictors of COVID-19 mortality: A strong prediction with the decision trees. Heliyon, e14015. https://doi.org/10.1016/j.heliyon.2023.e14015.

Response. We thank the reviewer for this helpful critique and the introduction has been revised to include key findings from the 3 references provided. “Biological variables..” has been moved to the materials and methods section.  Main purpose for the study has been clarified.

Material and Method Section

The parameters that were compared between the patient groups in this study should be mentioned here. In addition, which analyzes are used for statistical comparisons between groups and how the distributions are controlled should be written in the statistical analysis section. In the statistical analysis section, it should be stated what purpose the methods were used for (for example, what was the principal component analysis used for). Here, only Pearson correlation analysis was used for correlation analysis. It may be in places where semen should be used. Did all variables satisfy the assumption of normality? Which statistical package program was used? information should be given about them.

Response. The parameters that were compared are shown in figures. We used Welch’s t test to account for unequal group sizes and unequal variances, with pooled degrees of freedom given by the Welch–Satterthwaite equation. Data was transformed by the original authors where appropriate to ensure normality, e.g. cytokine measurements were log10-transformed. Supplementary Fig. 7 was changed to use Spearman’s correlation instead of Pearson’s; differences in r and p values were marginal. Details of methods are already provided. For analysis, we used the Stars application from Aseesa Inc (https://aseesa.com/).

Result Section

What exactly does Figure 5 and Figure 6 show? What do the numbers in the figures show? I'm having a really hard time figuring out what the methods are used for. Did you interpret the correlation results as an effect here? Correlation analyzes provide information about the presence and amount of positive/negative relationship between variables. However, I think that it is necessary to perform a regression analysis and calculate the odd-ratio values of the variables in order to determine the effect of the variables on the severity of COVID-19.

Response. We apologize for not clarifying. The heatmaps in Fig. 5 and 6 show log2 fold changes in all immunological parameters that were significant in ICU versus non-ICU female and male patients. The same measures were then analyzed for variables such as DFSO, clinical scores etc. Heatmaps provide a bird’s eye view of the changes allowing for simultaneous visualization of multiple measures rather than individual bar charts. Polynomial regression was carried out for select symbol combinations and the resulting curves are shown in correlation scatter plots, but our aim was to provide a general overview of changed/correlated measures rather than predicting outcomes such as severity, which would have exceeded the scope. This explanation has been added in the methods section.

Discussion and Conclusions section

The limitations of the study should be mentioned. In the conclusion part, information about the most important findings should be given without entering the comments.

Response. The sections have been revised as suggested. All changes are shown in red text.